# Gender Effects of Dioecious Plant *Populus cathayana* on Fungal Community and Mycorrhizal Distribution at Different Arid Zones in Qinghai, China

**DOI:** 10.3390/microorganisms11020270

**Published:** 2023-01-19

**Authors:** Zhen Li, Na Wu, Ting Liu, Ming Tang

**Affiliations:** 1Institute of Applied Biotechnology, College of Agriculture and Life Science, Shanxi Datong University, Datong 037009, China; 2State Key Laboratory for Conservation and Utilization of Subtropical Agro-Bioresources, Guangdong Key Laboratory for Innovative Development and Utilization of Forest Plant Germplasm, College of Forestry and Landscape Architecture, South China Agricultural University, Guangzhou 510642, China; 3College of Biology and Agriculture, Zunyi Normal College, Zunyi 563000, China

**Keywords:** fungal community, poplar, males and females, mycorrhizal fungi, drought

## Abstract

Dioecious plants have a wide distribution in nature and gender effect may cause significant alterations in rhizosphere fungal community and soil properties. However, little is known regarding changes in response to dioecious plants. This study aimed to investigate the effects that the dioecious plant, *Populus cathayana*, and regions of different arid levels have on the fungal community, mycorrhizal distribution, soil enzymatic activities, and nutrient contents. This study characterized fungal and soil factors from the rhizosphere of the dioecious plant *Populus cathayana* located in the semi-humid regions (Chengguan), semi-arid regions (Sining, Haiyan) and arid regions (Ulan, Chaka). Rhizosphere soil was collected from each site and gender, and the total fungal genomic DNA was extracted. DNA amplicons from fungal ITS region were generated and subjected to Illumina Miseq sequencing. A total of 5 phyla, 28 classes, 92 orders, 170 families, and 380 genuses were observed. AMF distribution peaked at Chaka, which did not conform to the trend. Gender had significant effects on fungal communities: there were obvious differences in fungal OTUs between genders. Alpha diversity raised at first and then decreased. RDA results showed available P, available K, pH, ALP activity, ammonium N, EC, water content and catalase activity were the key contributors in sample areas. Our results suggested potential interaction effects between plant gender and fungal community.

## 1. Introduction

Drought is an abiotic environmental stress, and seriously limits plant growth at different levels [1,2]. With historical records of precipitation, streamflow, and drought indices, increases in aridity is observed in recent years [3,4]. Furthermore, in view of climatic variation models, researchers suggest that in many areas of the world, plant production losses and ecological damages due to increasing water shortage will further aggravate its impacts [2,5]. To overcome the ill effect of drought and keep the balance in the gender ratio, researchers have tested and selected plants and fungi through conventional techniques [6]. In this case, a more pragmatic approach is to find a suitable symbiotic association of microbes under drought conditions to enhance the growth of hosts.

With more and more harsh environments and increasing demand for energy and industrial plants, the considerably better choice is selecting and growing native plants in abiotic stress regions. Situated in the northwest of China, Qinghai province is severely suffered from drought. As well, the dry type here belongs to the “Agricultural Drought”, characterized by below-average precipitation, intense but less frequent rain event, and above-normal evaporation, resulting in reduced agricultural production and plant growth [3,4]. Widely distributed in Qinghai province, northwest China, the native tree species *Populus cathayana* is generally used in drought regions. As a dioecious plant and widespread in this region, *P*. *cathayana* is important in restoring the ecosystem [6,7,8].

Adaptation of plants to drought is the result of multiple events, which give rise to accumulations in growth and physio-biochemical processes, such as changes in structure, photosynthesis, and water uptake [6,9]. Apart from internal events, external interactions with microbes are also important. Fungi are one of the most diverse groups of organisms and are important in ecosystem processes [10,11]. With governing soil carbon cycling, plant pathology, and nutrition, fungi are integral ecosystem agents [12]. Although it has been widely known that fungal communities in the specific ecosystem are not a random combination of species, researchers still seek to understand the process of shaping community assembly [13,14].

Specific instances of male and female to stress are well-reported and thoroughly studied [6,15], however, the differences in rhizosphere soil conditions and fungal communities of different genders in arid regions are still poorly understood. In previous studies, male plants performed better than females under kinds of abiotic stresses [16,17]. As well, greater reproductive expenditure in females is considered one of the primary drivers of male-biased ratios in plant populations [18]. In our previous studies, we found that *P*. *cathayana* males and females differed in preference for arbuscular mycorrhial (AM) symbiosis, which suggested the potential effect of microbe on plant gender ratio and different drought-tolerance mechanisms on different genders [6,18]. In-depth studies were needed. We designed this experiment to explain the interaction between dioecious plants and fungal communities in semi-arid and arid regions.

The main objective of this study was to test the effects of drought and gender on the dynamics of soil nutrients, soil enzyme activity, mycorrhizal fungal distribution, and fungal community functionality and structure. Furthermore, we tried to test the main soil factors that dominated the fungal distribution. Because of the above review, we hypothesized that drought-diminishing annual rainfall would limit soil enzyme activity, modify fungal community and diversity and restrict the richness of mycorrhizal fungi. To test this hypothesis, a manipulative experiment was set in which the annual precipitation and rainfall patterns were selected in a fully replicated set of sampling sites to simulate gradient levels of annual drought in the Qinghai Plateau, northwest of China.

## 2. Materials and Methods

### 2.1. Site Description and Soil Sampling

Sample sites were set in Qinghai province, China, a typical drought-affected region, as shown in Figure 1. With above 2000 m in altitude, little precipitation, and much evaporation, Qinghai province, located in northwest part of China, becomes one of the most ecologically fragile regions in the world. With an average annual temperature of 2.3 °C, average annual precipitation of above 300 mm, and average annual sunshine hours of over 3000 h, the climate shows a typical decrease in precipitation from east to west and from south to north. The main vegetation type and main soil type there are grassland and steppe soil.

In view of the distribution of water, 5 sampling sites with gradually decreasing annual precipitation, including Chengguan (S1), Sining (S2), Haiyan (S3), Ulan (S4) and Chaka (S5) were selected. Their geographical and climatic conditions are listed in Table 1. Chaka is famous for its landscape, salt lake, which is reflected in higher soil salt concentrate. The climate at 5 sample sites belongs to the typical plateau continental climate, with average annual rainfalls ranging from 520 to 159 mm. As well, all sample sites are located at Qinghai plateau, with elevations ranging from 2271 to 3108 m above sea level.

Samples were conducted in October 2016 with 3 plots (20 × 20 m^2^) at each site. In each plot, we selected 5 healthy, 20-year-old *P*. *cathayana* males and females respectively, and collected the roots and rhizospheric soil from a layer of 0~45 cm in depth (L1 refers to 0~15 cm; L2 refers to 15~30 cm; L3 refers to 30~45 cm) after moving the upper 5 cm of top-soil. The roots and soil samples from the same gender, same layer, and in the same plot were pooled as composite root and soil samples. Part of the root samples was stored in formaldehyde-acetic acid alcohol (FAA) for mycorrhizal fungal infection measurement [19]. Soil samples were stored at −20 °C for the measurement of soil properties and fungal community.

### 2.2. Soil Properties Measurement

The soil sample was weighed as fresh weight (FW), and oven-dried at 80 °C until constant weight (CW). The soil water content (%) was calculated as (FW-CW)/CW × 100. A pHS-3D digital pH meter (Leici PHS-3D, Shanghai, China) with a combined glass-calomel electrode was used for soil pH (1:5 soil-to-water solution). The total soil organic carbon (SOC) was measured with the dry combustion method (total organic carbon analyzer, TOC-VCPH, Shimadzu, Japan) as described by Xu et al. [20]. Nitrate N and ammonium N were detected with a continuous-flow Autoanalyzer (BRAN+LUEBBE AA3, Germany) [21].

### 2.3. Mycorrhizal Infection and GRSP Assessment

Before being cut into 1 cm fragments, roots, stored in FAA, were rinsed with water gently. According to the method described by Voets et al. [22], all root fragments were stained. One hundred and fifty randomly selected root fragments were placed on microscope slides and detected under a binocular optical microscope (Olympus Bx51, Tokyo, Japan) for AMF, EcMF, and DSE infections, respectively. Infection rates were estimated with the method of Declerck et al. [23].

GRSP, a glycoprotein with N-linked oligosaccharides, was considered to be copiously produced by AMF [24]. Total glomalin and easily extractable glomalin were extracted according to the method of Wright and Upadhyaya [24].

### 2.4. DNA Extraction and Sequencing

Soil samples stored at −20 °C were used for DNA extraction with the E.Z.N.A. Soil DNA Kit (Omega Bio-tek, Inc., Norcross, GA, USA) according to the protocols of the manufacturer’s instructions. To identification, the ITS1 V4 region with primer ITS1F (5′-CTTGGTCATTTAGAGGAAGTAA-3′) and ITS2 (5′-GCTGCGTTCTTCATCGATGC-3) was used [25]. PCR amplification of ITS amplification, Illumina MiSeq sequencing, and data analysis were performed by Personalbio Biotech Co., Ltd. (Shanghai, China).

### 2.5. Statistical Analysis

To determine the significance of the effects of site and gender on environmental factors, the statistical software package SPSS 26.0 (SPSS Inc., Chicago, IL, USA) was performed in this study. As well, these data were calculated with two-way analyses of variance (ANOVA) and the means were compared by Duncan’s multiple-range teats (*p* < 0.05). We compared the similarities in fungal communities from 2 genders and 5 sample sites based on the relative abundance data with a non-metric multidimensional scaling (nMDS). The data for nMDS was log transformed before and built up into a Bray-Curtis similarity matrix. These relative abundances were compared by one-way non-parametric Analysis of Similarities (ANOSIM) applied to gender and site. The RELATE analyses between fungal community and physical-distance was performed. We performed MDS, ANOSIM and RELATE analysis with multivariate statistical software for ecologist, Primer 7 (PRIMER-E Ltd., Albany, New Zealand). With the data from cooperative corporation, according to the results of detrended correspondence analyses (DCA), redundancy analyses (RDA) were used to determine the complex relationship between fungal community and environmental factors with Canoco 4.5 software (Biometry Centre, Wageningen, Netherlands).

## 3. Results

### 3.1. Distribution of Soil Elements

We measured soil available P, available K, nitrate N and ammonium N and SOC contents of rhizosphere of *P*. *cathayana* males and females from five different sites. The soil element result was listed in Table 2, suggesting that available P, available K, nitrate N and ammonium N contents had the same trend with the precipitation between sites: they decreased with drought level increase. Sample effects were obvious on soil elements distribution (*p* < 0.01).

Besides, compared to males’ rhizosphere (*p* < 0.01), the contents of available P were significantly higher in females’ (Table 2). As well, available K, available P, nitrate nitrogen and SOC contents were significantly affected by gender (*p* < 0.05). As well, two-way ANOVA showed that the interaction between samples and sex had significant effects on the elemental distributions (*p* < 0.01).

### 3.2. Soil Chemical Properties

To better understand the effects of soil factors, we studies soil EC, pH, water content and five soil enzymes from all sites. ECs from different sites showed gradually increase with the drought level (Table 2). As well, ECs at S5 were greatly higher than those from other four sites. pH was obviously affected by site, not by gender. In view of soil enzymes, the differences between genders were detected in all the five enzymes. The sucrose, catalase and dehydrogenase activities showed gradually decrease from S1 to S5 (except S2 to S3). Meanwhile, site had significant impacts on all five enzymes (*p* < 0.01), and so did gender (*p* < 0.01; except on the activity of catalase and ALP enzymes). With two-way ANOVA, we detected that the interaction of sites and genders significantly affected (*p* < 0.01).

### 3.3. Mycorrhizal Status and Glomalin Contents

The abundance and distribution of three mycorrhizal fungi were differently affected by gender (Table 3). AMF colonization rate and spore density showed positive correlations with annual precipitation from S1 to S4, but significantly larger amount of AM distribution was detected in S5. We suggested this might be due to the high EC in S5. The values of EcMF and DSE colonization fluctuated greatly between sites, but differed obviously between genders. All the mycorrhizal distributions were significantly different between sampling sites (*p* < 0.01), and were obvious affected (*p* < 0.01) by gender (except AMF colonization rate). Two-way ANOVA showed that the interaction of site and gender had a significantly impact on mycorrhizal colonization rates and AMF spore density (*p* < 0.01).

Gender also has various effects on the contents of EEG, TG, and their ratios to SOC. EEG contents showed negative trend with increase of annual precipitation in different sampling sites. However, TG contents had slight fluctuation in five sites. The EEG, TG, EEG/SOC and TG/SOC showed significant differences between different sampling sites (*p* < 0.01) and genders (*p* < 0.01; except in TG content). As well, two-way ANOVA results showed that, EEG, TG and EEG/SOC were significantly affected by the interaction of genders and sites (*p* < 0.05).

### 3.4. Fungal Taxon Identification and Alpha Diversity

In the total samples of *P*. *cathayana* from five different sites, five phyla, including Zygomycota, Ascomycota, Basidiomycota, Glomeromycota, and Chytridiomycota were detected in all treatments. The proportion of Ascomycota increased from S1 to S5, illustrating their potential drought tolerance (Figure 2). Besides, in total, 28 classes, 92 orders, 170 families, and 380 genuses were observed.

In the total representative samples of *P*. *cathayana* males and females from five different sites, we obtained 1,910,559 sequences (79.35% of the total trimmed 2,407,716) and grouped them into 380 Operational Taxonomic Units (OTUs) at 97% similarity cut-off level. Rarefaction analysis (not shown) showed that with the increased number of measured sequences, the OTUs reached stable values, which indicates that most fungal sequences obtained in this study reflected the abundance and diversity of the fungi. Besides, related analyses between fungal OTUs and physical distance showed that the sample statistic (Rho) and its significance level were 0.024 and 39.3%, which suggested no significant relationship existed between them. In view of gender, 142 OTUs were only available in the male’s rhizosphere and 127 OTUs only existed in the female’s rhizosphere. Meanwhile, 111 OTUs were distributed no matter which gender (Figure 3A). Apart from gender, site effects existed in fungal distribution, 51, 60, 73, 60, and 43 OTUs existed only in S1, S2, S3, S4, and S5. As well, 42 OTUs were distributed in all sampling sites (Figure 3B).

Alpaha diversity was estimated by ACE, Chao1, Shannon, and Simpson (Table 4). Obvious regulars of fungal diversity of both genders with annual precipitation are detected: the diversity decreased at first and increased from S3, but the rise phase was slighter in female plants than that in males. The differences in the fungal community were compared based on the relative abundances with nMDS (Figure 4). From the point of fungal community structure, the nMDS similarities between genders of the same site were above 90% (except seedlings of S1), meanwhile suggesting that the site can better distinguish the fungal community than gender.

### 3.5. RDA Results

RDA results showed that more than 54.6% of the variance in fungal communities could be annotated. RDA 1 and RDA 2 accounted for 34.8% and 19.8% of the variance, respectively (Figure 5). Available P (*p*-value, 0.014), available K (0.024), pH (0.04), ALP activity (0.108), ammonium N (0.016), EC (0.026), water content (0.008) and catalase activity (0.004) were the key contributors in this analyze. Fungal communities in two genders of five sites were distinctly clustered into four groups: (1) S1M, S1F, S4M and S4F; (2) S2F, S2M and S3M; (3) S3F; (4) S5M and S5F.

*Aspergillus flavus*, *Cortinarius* sp., *Mortierella alpine*, *Mortierella polygonia*, *Pyronemataceae* sp. and *Rhizophydium aestuarii* mainly distributed in group 1, and showed a positive correlation with EC, and negative with pH and ALP activity. *Mortierella* sp., *Chytridiomycota* sp., *Mortierella horticola*, *Helotiales* sp., *Tetracladium maxilliforme*, *Ascomycota* sp., *Tremellomycetes* sp., *Phoma sclerotioides*, *Claroideoglomus claroideum*, *Podospora* sp., *Podospora vesticola*, *Humicola nigrescens*, *Leotiomycete* sp. and *Rhizophagus intraradices* mainly distributed in group 2, which were positive with water content and negative with available P, available K and ammonium N. *Microascales* sp., *Thelebolus globosus* and *Fusarium oxysporum* mainly distributed in group 3, which were positive with pH and ALP activity. *Emericellopsis* sp., *Rhizophydium littoreum*, *Hypocreales* sp., *Cladosporium* sp., *Stagonosporopsis ajacis*, *Alternaria destruens*, *Sordariomycetes* sp., *Pezizomycetes* sp. mainly distributed in group 4, which were positive with available P, available K, ammonium N, and catalase activity. Besides, the soil conditions and fungal communities in S1 were similar to S4, and the same was in S2 and S3 (Figure 5). As well, we detected that, compared with gender, the effect of sites was more effective, which was in line with the results of nMDS. Besides this one similarity, the distinction between sites, such as S2 and S3, S1 and S4, was also similar.

## 4. Discussion

Climate factors, soil, and spatial patterns are the most efficient predictors for edaphic fungal richness and community structure at a global scale [12]. During these factors, soil properties showed several trends with climatic differences [26]. These results suggested the potential role of climate in altering edaphic conditions. Fu et al. [12] said that, climate could affect soil properties, which meanwhile affect the fungal status. Our results showed that poplar plant gender and environmental arid level differently affected root colonization, soil properties, fungal community structure and mycorrhizal fungi in the underlying soil in northwest China.

In this study, due to climatic reason, drought level increased from S1 to S5, which might be the main cause to spatial variation in soil properties and fungal diversity and richness between sites. The decrease in soil enzyme activities with high-severity drought was likely due to decreased plant metabolism. At drought regions, due to lower water uptake and higher soil infiltration rates, host plants and soil microbes maintain low-level metabolism to survive, which means lower activity and less exudates. This resulted in decreased soil elements contents. So, in line with this trend, in our study, contents of soil elements decreased with increased climatic drought level.

Previous studies suggested the importance of plant species on soil properties and microbes, resulting in different soil properties from rhizosphere of different plant species grown in the same substrate [8,27]. However, in the same plant species, gender effects were always ignored, especially in field experiments. In some cases, soil properties of the rhizosphere may be further influenced by plant gender. With more and more theoretical support, researchers believed that this effect really existed and was always ignored in previous studies [6,7]. Thus, in this study, we focused on the impacts of this potential factor on soil properties and fungal status, attempting to explain its importance.

The results showed that gender increased soil heterogeneity. At same site, soil properties always differed in the rhizosphere of male and females. Soil elements (except ammonium N content) were significantly affected by gender, which revealed the existing role of gender effect. As well, among the activities of sucrase, dehydrogenase, and urease, a significant gender effect was also detected. Specific instances of male, and female to stress are well-known and thoroughly studied [6,17]. Due to different responses of genders to drought, the rhizosphere of male and female host plants differed, which directly resulted in the different soil properties. Besides, impacted microbes by gender might be another cause of soil heterogeneity.

Plants always survive stresses by altering their morphology and physiology and developing some mechanisms [28]. Beyond this, soil microbes, developing symbiotic and non-symbiotic associations with hosts, are able to make plants survive under stress. Among the most significant components of soil microbes, mycorrhizal fungi are a link from underground to aboveground [29,30]. To explore the zone beyond the basic one formed around the roots, plants formed mycorrhizal symbiosis which extends their external hypha to increase the volume of soil the host can reach [31]. They interact with hosts forming a complex relationship, where fungi and plants receive benefits from each other. Those mycorrhizal fungi were reported to be helpful to hosts to survive under drought. Generally, AMF root colonization under drought stress was found to be lower compared with the patterns detected under well-watered conditions [32]. However, reverse results were obtained in a variety of research [6,33]. However, the inoculation rate was not the determinant criterion in measuring the key component of AMF symbiosis in helping plants to cope with adverse environments. Even with less than 10% root colonization, AMF inoculation alleviated drought stress [12,33]. In this research, AMF inoculation and spore density decreased with the decrease of annual precipitation from S1 to S4. But at the most drought site, AMF inoculation rates were higher than those at S3 and S4, which may be due to the high salt content in S5. Besides, AMF root inoculation in males was found to be not lower than those in females, which illustrated potential different mechanisms of AMF in forming a symbiosis with different genders of dioecious plants.

EcMF and DSE inoculation rates varied erratically between sites and genders. As well, among the three kinds of mycorrhizal fungi, the distribution of DSE was the least in popular roots. This may be due to no external hypha in DSE species [34]. AMF inoculation rate and spore density decreased from S1 to S4, which resulted from the decreased annual precipitation, but rose at S5, which may be due to the irregularly high EC in S5. Among these mycorrhizal fungi, site and gender had significant effects on colonization. Wu et al. [15] suggested that there was no difference in mycorrhizal colonization between genders in the pot experiment, which was not similar to our results.

Glomalin was copious production of glycoprotein by AMF and good for maintaining fine water infiltration rates and adequate aeration for host growth [24]. In this study, both TG and EEG were detected high contents at all sites, which suggested the important role of AMF in keeping fine water infiltration in drought areas. Although the AMF inoculation rate showed a decreased trend with the increase of drought level in sites, glomalin contents showed a reverse trend, which may be due to the stimulation by drought and the low activity of soil microbes. EEG contents showed a negative trend with an increase in annual precipitation in different sampling sites. However, TG contents had a slight fluctuation in five sites. But the ratio of glomalin to SOC content did not fluctuate greatly between sites. These results suggested the significant effect of glomalin in keeping fine soil status and great contribution to SOC accumulation.

At the global scale, edaphic microbial richness and community structure were severely affected by environments, such as climate factors, soil, and spatial patterns [12]. One of the most important factors in determining the microbial diversity and community in the rhizosphere is plant species, resulting in different microbe compositions for various species grown in the same soil [35,36]. The importance of soil properties and plant species was always discussed in the microbial community [7,8,37]. But in some cases, host gender may have a further influence on microbial composition than soil and plant species.

Drought has direct and indirect effects on fungal community. Drought, as a major controlling factor of the soil microbial community, might directly limit its activity, which depends on the duration and intensity [38,39]. Besides, some plant species, adapting poorly to drought, may disappear, which influences the fungal community indirectly [40]. Plant diversity always had positive effects on microbial diversity. As shown in nMDS results, nMDS similarity levels at the same site (except seedlings at S1) were over 90%, which reveals the greater effect of the site than gender on the fungal community. Meanwhile, fungal communities from all treatments showed over 80% nMDS similarity. The differences between sites resulted from different environmental factors, rather than physic distance, which was shown by related analyses results. Besides, in this study, the diversity of the fungal community of both genders decreased at first and rose with the decrease in annual rainfall at different sites. This indicates that, in an arid region, the drought effects dominated the fungal diversity, independently of gender. However, this trend was weaker in females compared with males, which may depend on the gender differences in root exudates or stress response systems [6,18].

RDA results suggested the importance of available P, available K, ammonium N, pH, ALP activity, and EC on soil heterogeneity in this study, which was in line with several studies before [41]. Moreover, the size and diversity of fungal populations were mostly limited by the level of drought, and the amount of soil element sources available. This is consistent with our findings of diversity and richness results. High microbial diversity often resulted in high nutrients available, due to the enhanced microbial biomass turnover [8,37], which was in line with our findings of fungal diversity and soil element contents. Compared with nMDS results, the environmental data was added in RDA. These two analyses all suggested two similar conclusions: (1) compared with gender, the effect of site was more powerful; (2) biotic and abiotic environments of S2 and S3 were very similar, and the conditions of S5 was really distinct, which suggested the potential effect of climate [12].

However, detailed mechanism about how the environment and gender-modified soil properties and soil microbial community, which needs further study, especially on detail research on every part. In addition, how to efficiently utilize microbes to improve the ecological status, especially under drought stress, still needs more attempts.

## 5. Conclusions

This study distinguished the effect of different arid conditions on fungal communities. As well, the results clearly showed that there were interactions between the host plant gender and all the indicators we detected. Soil properties are generally affected by arid conditions, but not all by gender. Meanwhile, the potential preference of fungal community for two genders was detected. With a former study on the effects of AMF on plant physiology under drought conditions, symbiotic fungi may have the potential ability to ecological stability and species continuation of *P*. *cathayana*. Regarding plant gender ratio and fungal community properties, more attention should be paid to their potential associations.

## Figures and Tables

**Figure 1 microorganisms-11-00270-f001:**
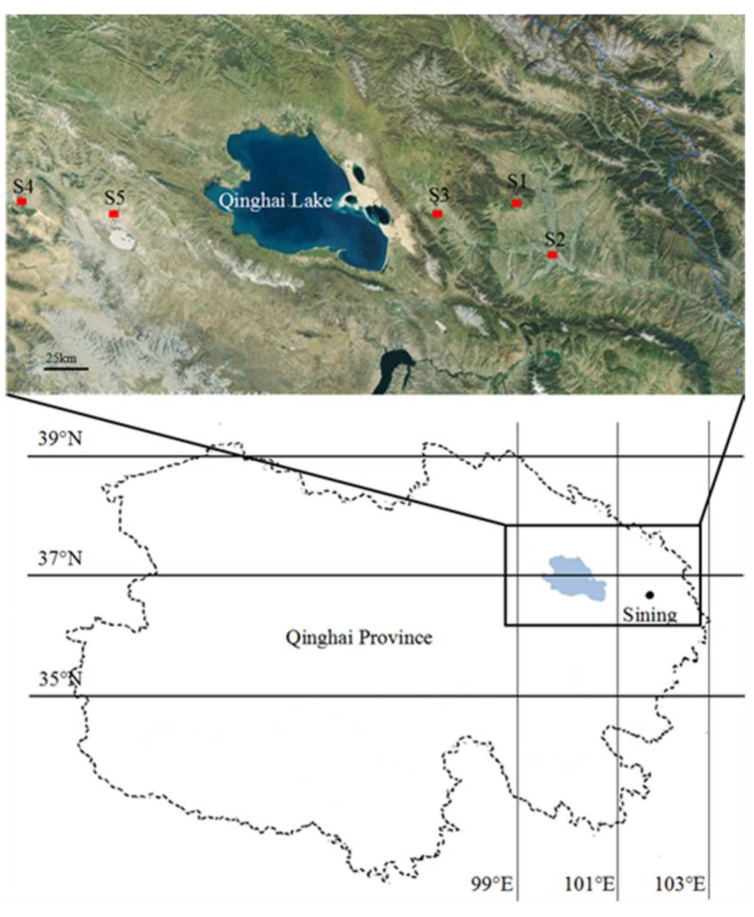
Distribution of 5 study sites.

**Figure 2 microorganisms-11-00270-f002:**
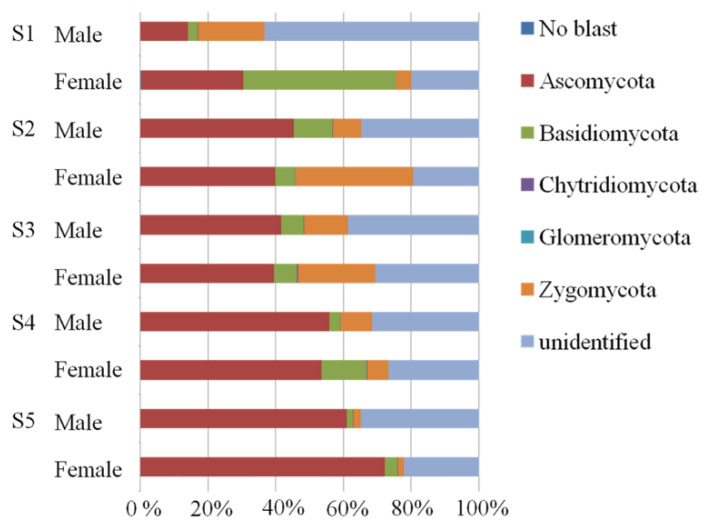
Proportions of different fungal phyla at five study sites.

**Figure 3 microorganisms-11-00270-f003:**
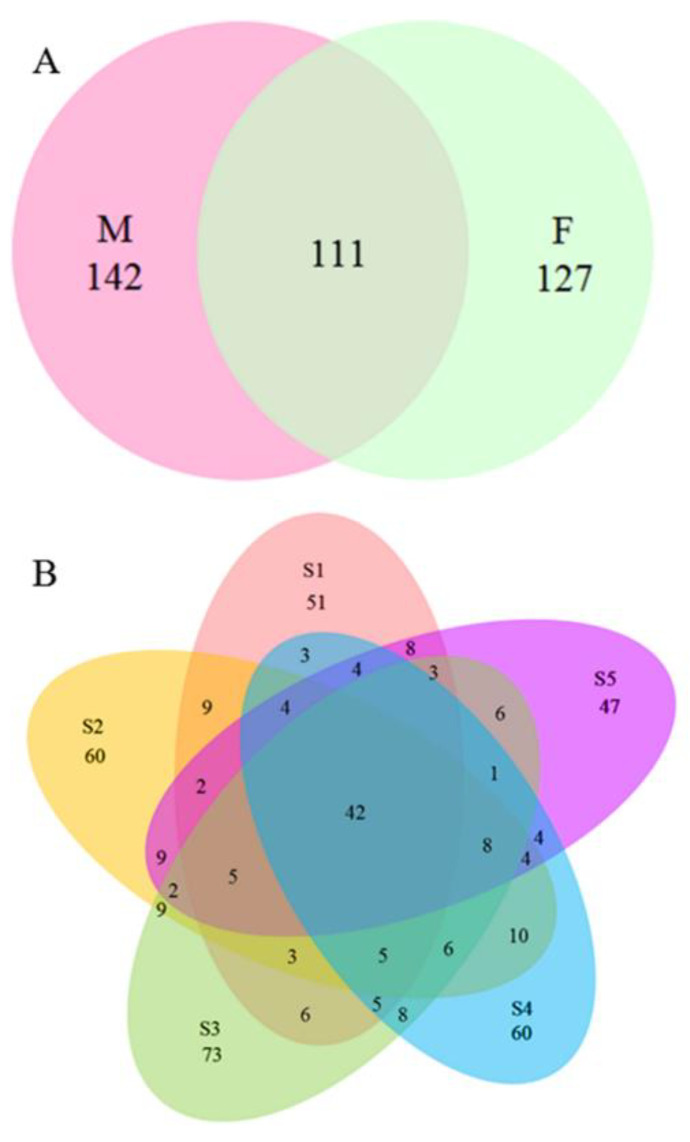
Proportions of OTUs between genders (**A**) and sites (**B**). M: male; F: Female.

**Figure 4 microorganisms-11-00270-f004:**
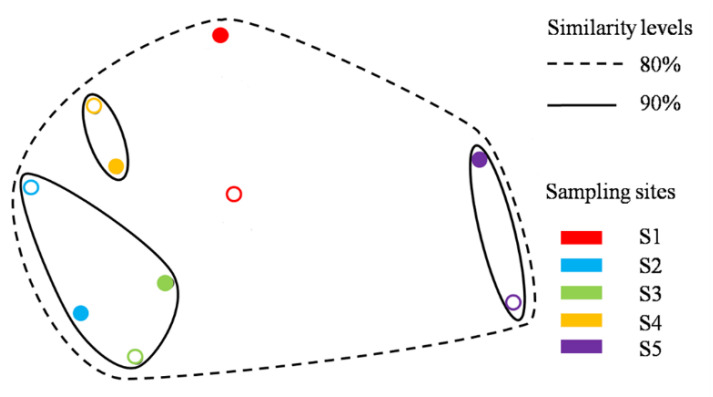
nMDS results of fungal community similarity between genders and treatments. Different circles represent different treatments. Solid and hollow circles represent male and female, respectively. Different colors mean different sites.

**Figure 5 microorganisms-11-00270-f005:**
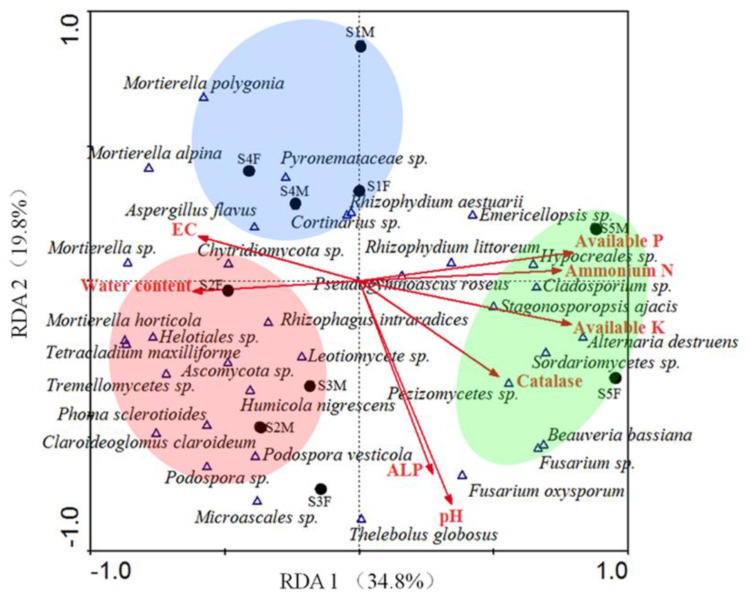
RDA results. Solid circles mean sites. Blue hollow triangles mean fungal species. Red arrows mean soil properties.

**Table 1 microorganisms-11-00270-t001:** Description of the study sites.

Sampling Sites		Longitude/Latitude	Altitude/m	Annual Precipitation/mm
S1	Chengguan (CG)	101°30′51″ E/37°2′49″ N	2644	520
S2	Sining (SN)	101°40′50″ E/36°39′19″ N	2271	390
S3	Haiyan (HY)	101°0′18″ E/36°53′46″ N	3004	380
S4	Ulan (UL)	98°27′59″ E/36°57′10″ N	3013	159
S5	Chaka (CK)	99°4′24″ E/36°47′31″ N	3108	193
S5	Chaka (CK)	99°4′24″ E/36°47′31″ N	3108	193

**Table 2 microorganisms-11-00270-t002:** Soil properties in rhizosphere of host plants at different sites.

Sample Sites	Sex	Available P (g·kg^−1^)	Available K (g·kg^−1^)	NO3-N (g·kg^−1^)	NH4-N (g·kg^−1^)	EC dmS·m^−1^	pH	Water Content (%)	SOC (g·kg^−1^)	ALP (10^−2^ mg·g^−1^·h^−1^)	Catalase (mg·g^−1^·h^−1^)	Sucrase (mg·g^−1^·h^−1^)	Dehydrogenase (µg·g^−1^·h^−1^)	Urease (10^−2^ mg·g^−1^·h^−1^)
S1	Male	0.30 b	504.10 b	0.61 a	0.98 b	425.69 f	8.33 b	6.62 e	32.74 b	0.90 c	7.79 a	1.46 a	0.29 a	0.66 cd
	Female	0.32 a	553.89 a	0.51 bc	1.14 a	472.66 e	8.58 a	5.35 f	31.36 cd	0.99 b	6.86 b	1.23 c	0.27 b	0.65 cd
S2	Male	0.25 d	247.98 d	0.47 de	0.83 d	537.32 d	7.99 c	10.37 ab	29.80 e	0.42 g	4.79 f	1.26 c	0.28 b	0.68 cd
	Female	0.28 c	232.18 d	0.40 h	0.90 c	609.40 c	7.96 c	5.07 f	23.92 g	0.81 de	3.89 g	1.25 c	0.12 g	0.69 cd
S3	Male	0.22 f	352.14 c	0.53 b	0.88 c	563.48 d	8.35 b	10.71 a	35.27 a	1.09 a	6.77 b	1.35 b	0.30 a	0.95 b
	Female	0.23 e	337.30 c	0.49 cd	0.88 c	563.98 d	8.22 b	9.61 bc	32.57 bc	0.81 de	6.41 c	1.36 b	0.15 c	0.69 cd
S4	Male	0.17 hi	207.85 e	0.41 gh	0.73 e	626.63 c	8.35 b	11.14 a	24.13 g	0.72 e	5.24 d	1.07 d	0.12 e	0.95 b
	Female	0.20 g	209.62 e	0.44 ef	0.67 f	697.16 b	8.62 a	7.61 d	23.29 g	0.87 cd	5.14 de	1.02 d	0.10 f	0.64 cd
S5	Male	0.16 i	207.84 e	0.44 efg	0.77 e	924.82 a	7.96 c	9.30 c	30.26 de	0.85 cd	4.80 f	1.10 d	0.13 d	0.61 d
	Female	0.18 h	241.26 d	0.42 fgh	0.75 e	958.68 a	8.21 b	7.76 d	27.24 f	0.56 f	4.89 ef	1.39 ab	0.10 f	0.72 c
*F* _Sample_		105.30	224.28	190.06	390.57	90.99	9.15	170.73	409.04	127.41	860.85	549.27	241.3	196.96
*P* _Sample_		0.00 **	0.00 **	0.00 **	0.00 **	0.00 **	0.02 *	0.00 **	0.00 **	0.00 **	0.00 **	0.00 **	0.00 **	0.00 **
*F* _Sex_		186.92	1.85	116.06	1.76	66.03	3.69	536.72	23.57	1.14	1.32	399.25	320.46	663.41
*P* _Sex_		0.00 **	0.04 *	0.00 **	0.14 ^NS^	0.00 **	0.08 ^NS^	0.00 **	0.00 **	0.292 ^NS^	0.12 ^NS^	0.00 **	0.00 **	0.00 **
*F* _Sample×Sex_		7.45	27.29	32.62	38.75	5.75	16.80	54.92	2.38	122.68	29.18	186.15	572.04	153.73
*P* _Sample×Sex_		0.01 **	0.00 **	0.00 **	0.00 **	0.01 *	0.00 **	0.00 **	0.08 ^NS^	0.00 **	0.00 **	0.00 **	0.00 **	0.00 **

Note: Different letters of each column indicate significant differences between treatments (*p* < 0.05). *: significant effect at 0.01 < *p* < 0.05; **: significant effect at *p* < 0.01; ^NS^: no significant effect.

**Table 3 microorganisms-11-00270-t003:** Mycorrhizal distribution and glomalin contents in the rhizosphere of host plants at different sites.

Sample Sites	Sex	Colonization Rate (%)			Spore Density ((10g)^−1^)	EEG (g·kg^−1^)	TG (g·kg^−1^)	EEG/SOC (%)	TG/SOC (%)
		AMF	EcMF	DSE					
S1	Male	50.79 a	27.90 fa	11.72 cd	179.94 c	0.88 c	4.78 de	2.96 d	16.80 cd
	Female	51.50 a	57.85 c	12.04 c	166.18 d	0.91 bc	4.98 cd	3.42 b	19.41 b
S2	Male	49.65 a	34.74 d	12.38 bc	164.36 d	0.97 b	5.18 bc	2.94 d	13.22 g
	Female	42.03 b	83.88 a	7.13 f	166.64 d	1.21 a	5.50 a	3.79 a	14.99 ef
S3	Male	36.83 c	31.39 e	13.07 b	92.79 f	0.97 b	3.97 g	3.10 cd	16.22 de
	Female	33.74 c	18.54 h	11.05 de	101.69 e	0.97 b	4.42 f	3.28 bc	21.15 a
S4	Male	26.87 d	63.15 b	14.91 a	77.19 g	1.22 a	4.51 ef	2.13 f	13.97 fg
	Female	23.04 e	23.66 g	10.83 e	67.91 g	1.27 a	4.26 fg	2.53 e	14.43 fg
S5	Male	49.51 a	31.43 e	4.00 g	237.94 a	1.22 a	5.37 ab	3.47 b	16.67 cd
	Female	44.58 b	24.40 g	7.10 f	222.14 b	1.25 a	5.24 abc	3.79 a	17.77 c
*F* _Sample_		443.83	186.77	633.48	165.53	251.64	130.81	257.74	137.17
*P* _Sample_		0.00 **	0.00 **	0.00 **	0.00 **	0.00 **	0.00 **	0.00 **	0.00 **
*F* _Sex_		6.92	190.13	227.74	15.89	65.07	2.57	266.31	171.79
*P* _Sex_		0.06 ^NS^	0.00 **	0.00 **	0.00 **	0.00 **	0.06 ^NS^	0.00 **	0.00 **
*F* _Sample×Sex_		9.011	305.82	205.77	11.86	24.92	10.83	17.29	2.19
*P* _Sample×Sex_		0.01 **	0.00 **	0.00 **	0.00 **	0.00 **	0.00 **	0.00 **	0.12 ^NS^

Note: Different letters of each column indicate significant differences between treatments (*p* < 0.05). **: significant effect at *p* < 0.01; ^NS^: no significant effect.

**Table 4 microorganisms-11-00270-t004:** Comparison of diversity indices among different sites.

Gender	Site	Chao Diversity	ACE Diversity	Simpson Diversity	Shannon Diversity
Male	S1	800.04 d	811.26 c	0.176 a	2.70 c
	S2	903.44 ab	920.70 a	0.072 bc	3.93 a
	S3	951.84 a	918.96 a	0.045 c	4.05 a
	S4	885.91 bc	871.61 ab	0.085 bc	3.78 ab
	S5	875.80 cd	863.059 bc	0.108 b	3.24 bc
Female	S1	869.43 C	848.32 B	0.209 A	3.01 B
	S2	911.79 B	856.30 B	0.072 B	3.67 A
	S3	1002.12 A	1002.04 A	0.051 B	3.79 A
	S4	855.14 BC	952.83 A	0.053 B	3.87 A
	S5	904.85 B	890.92 B	0.068 B	3.63 A

Note: Different letters of each column indicate significant differences between sites (*p* < 0.05).

## Data Availability

Data is unavailable due to ethical restrictions.

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
