# Peer review of "Gender Effects of Dioecious Plant *Populus cathayana* on Fungal Community and Mycorrhizal Distribution at Different Arid Zones in Qinghai, China"

_microorganisms, 2023, doi:10.3390/microorganisms11020270_

Round 1

Reviewer 1 Report

Authors have completed well research on the effects that dioecious Populous cathayana and drought levels have on the fungal community, mycorrhizal distribution, soil enzymatic activities, and nutrient contents, providing the potential interaction effects between plant sex and fungal community. The article is rich in content and clear in conclusion. However, there are some points to be clarified. Please find my comments. 

Comments:

L27 alpha diversity raised at first and then decreased? At which gradient?

L40 select plants and fungi, please introduce them more specifically.

L51 why mention the uses related to industrial here?

L68 please add a reference.

Table1 for the site Sining and Chaka, compared with other sites, the altitude difference between these two is nearly 1000 m. Altitude is known to cause temperature changes, will this affect the interpretation of the results? This may need to be addressed in the discussion.

L132 here ITS region, while in the abstract, (L23) said V4 region, which needs to be clarified. 

Table 2 the result of soil properties needed to be represented as mean ± se, as should Table 3 and 4.

L171 from S2 to S3, the trend was not decreasing.

Figure2 it’s confused that what’s the difference between ‘No blast’ and unidentified, for more articles used ‘others’.

Figure 4 The results of the statistical analysis of NMDS need to be listed.

Page12 L139 for the discussion of results of fungal community, you should read the article ‘Sex-specific interactions shape root phenolics and rhizosphere microbial communities in Populus cathayana’ and make this section more complete.

Author Response

Authors have completed well research on the effects that dioecious Populous cathayana and drought levels have on the fungal community, mycorrhizal distribution, soil enzymatic activities, and nutrient contents, providing the potential interaction effects between plant sex and fungal community. The article is rich in content and clear in conclusion. However, there are some points to be clarified. Please find my comments. 

AnswerThanks for your affirmation!

Comments:

L27 alpha diversity raised at first and then decreased? At which gradient?

Answer: Thanks for your question! Alpha diversity raised and then decreased along annual precipitation from S1 to S5.

L40 select plants and fungi, please introduce them more specifically.

Answer: Thanks for your suggestion! We have revised this.

L51 why mention the uses related to industrial here?

Answer: Thanks for your suggestion! We have deleted this sentence.

L68 please add a reference.

Answer: Thanks for your advice! We have added reference here.

Table1 for the site Sining and Chaka, compared with other sites, the altitude difference between these two is nearly 1000 m. Altitude is known to cause temperature changes, will this affect the interpretation of the results? This may need to be addressed in the discussion.

Answer: Thank you for your suggestion! This is an important factor which we ignored before. And we have added this part in discussion.

L132 here ITS region, while in the abstract, (L23) said V4 region, which needs to be clarified. 

Answer: Thanks for your advice! We have added this as you suggested.

Table 2 the result of soil properties needed to be represented as mean ± se, as should Table 3 and 4.

Answer: Thank you for your suggestion! As you see, table 2, 3 and 4 were truly big table, there was not enough space adding “± se”. And we suggested to add raw data in afflication part.

L171 from S2 to S3, the trend was not decreasing.

Answer: Thank you for your question! We have revised this sentence.

Figure2 it’s confused that what’s the difference between ‘No blast’ and unidentified, for more articles used ‘others’.

Answer: Thank you for your question! Others means these sequences were still recognized as fungi, while “No blast” means these sequences were not blasted.

Figure 4 The results of the statistical analysis of NMDS need to be listed.

Answer: Thanks for your advice! We have added this as you suggested.

Page12 L139 for the discussion of results of fungal community, you should read the article ‘Sex-specific interactions shape root phenolics and rhizosphere microbial communities in Populus cathayana’ and make this section more complete.

Answer: Thanks for your advice! We have revised this part as you said.

Reviewer 2 Report

The authors propose a study on the effects of Populous cathayana, and regions of different arid levels have on fungal community, mycorrhizal distribution, soil enzymatic activities and nutrient contents.

The paper is mostly well written, however I have a few suggestions.

The introduction should give more details on Populous cathayana, to justify the chosen plant.

The authors should state the null hyothesis.

In the discussions section, the authors should stress the originality of the study.

The authors should state the limitations of the study and future directions.

Author Response

The authors propose a study on the effects of Populous cathayana, and regions of different arid levels have on fungal community, mycorrhizal distribution, soil enzymatic activities and nutrient contents.

The paper is mostly well written, however I have a few suggestions.

Answer: Thanks for your affirmation!

The introduction should give more details on Populous cathayana, to justify the chosen plant.

Answer: Thanks for your suggestion! We have revised this as you suggested.

The authors should state the null hyothesis.

Answer: Thanks for your suggestion!

In the discussions section, the authors should stress the originality of the study.

Answer: Thanks for your suggestion!

The authors should state the limitations of the study and future directions.

Answer: Thanks for your suggestion! These statements was already shown at the end of discussion part.

Reviewer 3 Report

In general the manuscript is well written, and the authors showed clearly how they archieved their goals. 

The manuscript suggest that the interaction effects between plant gender and fungal community could be realted, as they demonstrated in the results.

A review of English and grammar is necessary. 

  •  

Author Response

In general the manuscript is well written, and the authors showed clearly how they archieved their goals. 

The manuscript suggest that the interaction effects between plant gender and fungal community could be realted, as they demonstrated in the results.

A review of English and grammar is necessary. 

Answer: Thanks for your affirmation and suggestion! We have revised our paper as your suggestion.